# Study of Decomposition of Chemical Warfare Agents using Solid Decontamination Substances

**DOI:** 10.3390/toxics7040063

**Published:** 2019-12-07

**Authors:** Tomas Capoun, Jana Krykorkova

**Affiliations:** Ministry of Interior–General Directorate of the Fire Rescue Service CR, Population Protection Institute, Na Lužci 204, 533 41 Lázně Bohdaneč, Czech Republic; tomas.capoun@ioolb.izscr.cz

**Keywords:** chemical warfare agent, decontamination, decontamination sorbent, degradation efficiency, FTIR spectrometry, ATR technique

## Abstract

The decontamination of chemical warfare agents is important for the elimination or reduction of the effects of these substances on persons. Solid decontamination (degradation) sorbents that decompose dangerous substances belong among modern decontamination substances. The aim of the study was to design a procedure for monitoring the degradation of chemical warfare agents using such sorbents. The degradation of soman, VX [*O*-ethyl-*S*-(diisopropylaminoethyl)methylphosphonothioate] and sulphur mustard (chemical warfare agents) was monitored using FTIR spectrometry with the attenuated total reflection (ATR) technique. During the development and validation of this process, bonds were found in the substance molecule that decomposed and the positions of the absorbance bands corresponded to the vibration of these bonds. The evaluation of the degradation efficiency procedure for sorbents on chemical warfare agents was designed based on this study. We present the result of the measurements graphically as the time dependence of the distributed chemical warfare agent ratio, and the reaction times required to decompose 50% and 90% of the original amount of the substance.

## 1. Introduction

Decontamination of chemical warfare agents (CWAs) is a significant measure of active protection against the consequences of uncontrolled leakage of CWAs into the environment, or the consequences of their misuse. Decontamination is carried out in the event that human lives, health, properties, or the environment are at risk because of a contaminant’s hazardous properties, and to prevent the spread of CWAs or other hazardous substances. Decontamination may be carried out using chemical, physical, combined physical/chemical, mechanical and biological (biodegradation) methods [1,2]. Decontamination procedures that are combinations of different methods using chemical decomposition reactions are most frequently used [2,3].

Decontamination sorbents, also called degradation or destructive sorbents, represent a prospective development in decontamination substances. These substances absorb CWAs into the surface structure of the sorbent material, followed by chemical decomposition. Their widespread usability has been particularly recognised in the field of individual decontamination, which, in the case of human contamination, is a crucial measure for reducing the effects of CWAs on human beings [4,5]. The M291 Skin Decontamination Kit [3], with the synthetic sorbent Ambergard XE-555, has been used by the US military since the latter half of the 1980s. The market also offers a decontamination nano-sorbent, FAST-ACT, which is designed for the adsorption, neutralisation, and decontamination of a wide range of liquid and gaseous toxic substances, including CWAs [6,7]. Another means of decontamination with a degrading effect is the Polish Army IPP-95 kit [5]. Practical CWA decontamination experiments [8,9] have shown that not all decontamination sorbents react with CWAs to the desired degree of decomposition. In order to assess the usability of solids for CWA decontamination, it is therefore necessary to know their efficiency.

Regardless of the composition of the decontamination agent, two aspects of the effectiveness of the decontamination agent must be considered: decontamination efficiency and degradation efficiency.

The objective of assessing the decontamination efficiency is to determine the proportion of the initial CWA on the surface that the tested decontaminant is able to remove from the test surface under the test conditions [8,9,10]. First, the test surface is contaminated with drops of the CWA to the initial contamination density. After a certain period of exposure, the prescribed amount of decontaminant is applied to the surface. After a given period of time, the substance is removed from the surface (by dabbing off, rinsing with water, etc.). A sample of the surface is taken, and the total residual contamination caused by the CWA is determined using a selected analysis. This value in itself is a measure of decontamination efficiency and is usually compared with permissible residual contamination. Furthermore, the efficacy is expressed by the value of the so-called decontamination efficiency, which expresses the percentage of the initial amount of the contaminant removed from the surface.

The degradation efficiency means the degree of CWA decomposition using the decontamination agent. Degradation efficiency testing is simpler in the case of liquid decontamination mixtures, since it is a homogeneous reaction with a liquid CWA [10], while the reaction of a liquid CWA with a solid decontaminant is a heterogeneous reaction that requires a special procedure such as the one described in this paper. The conditions, efficiency and mechanism of CWA decontamination by metal-oxide-based solid sorbents have been discussed in previous studies [11,12,13,14]. The degradation of CWAs and their simulants have been monitored mainly during the development and comparison of decontamination sorbents. Solid-state NMR [15,16,17,18,19,20,21,22,23,24], GC/MS [15,20,25,26,27,28], GC/FID [27,29] methods and some FTIR techniques [17,25,27] predominate. In the Czech Republic, the methodology for evaluating the degradation efficiency of solid decontamination substances has, to date, been non-existent. Chemical laboratories of the Fire Rescue Service (FRS) must be able to assess the degradation efficiency of the decontamination products offered to FRS units. The method of testing efficiency must be implemented using the instruments with which these laboratories are equipped and which they will use.

Attention was focused on the infrared spectrometry method with Fourier-transform (FTIR spectrometry), together with the reflective measuring technique called ATR (attenuated total reflection). The procedure utilised infrared spectra of the CWA compounds with a decontamination sorbent, measured against the sorbent itself, to determine the degree of CWA decomposition based on the intensity of a particular absorbance band. The corrected area of the absorbance band was used to determine the bandwidth intensity, which was calculated using the OMNIC spectroscopic program [30]. Figure 1 shows the surface reading. The baseline points correspond with the limits of the range of wavenumbers.

The basic requirement for the use of FTIR spectrometry to study CWA decomposition is a knowledge of the bonds that decompose during the reaction, and the position of the absorbance bands characteristic to the vibrations of those bonds. Finding these bonds and bands was the main goal of this study.

## 2. Materials and Methods

The methodology was developed to monitor the decomposition of the CWAs *O*-(3,3-dimethyl-2-butyl)methylphosphonofluoridate (soman), 97% purity by potentiometric lanthanometric method using a fluoride-ion-selective electrode; *O*-ethyl-*S*-(diisopropylaminoethyl)methylphosphonothioate (VX), VOZ 072 Zemianske Kostolany, Slovakia, 82% purity by potentiometric thiomercurymetric method using a sulphide-ion-selective electrode); and bis(2-chlorethyl)sulphide (sulphur mustard), VOZ 072 Zemianske Kostolany, Slovakia, 98% purity by potentiometric argentometric method using a sulphide-ion-selective electrode.

The process of assessing degradation efficiency was verified for the FACT-ACT decontamination nanosorbent (NanoScale Corp., USA) and five samples of decontamination sorbents developed in the Czech Republic under the labels LB1, LB2, LB3, LB4 and LB5.

For the measurement, 0.10 g of tested decontamination sorbent was weighed into a 4 mL glass vial. A quantity of CWA corresponding to 0.02 g of pure substance was added to the sorbent via a Hamilton syringe and a timer was switched on. The mixture was thoroughly mixed in the vial with a spatula; a part was applied to the ATR accessory of the FTIR spectrometer. Within 0.5 min, the infrared spectrum, which corresponds to a time of 0 min, was measured. Furthermore, the spectrum of the mixture was measured at selected time intervals until the reaction time of 120 min was complete. As a background, the decontamination sorbent itself was measured by the FTIR method, using ATR.

The IR spectra were measured on a NICOLET iS10 FTIR spectrometer (Thermo Scientific, USA) with a single-reflection ATR accessory and diamond crystal (GladiATR Vision). Mixtures of sorbents with CWA were measured at the following parameters: number of scans: 32, spectral range: 4000–400 cm^−1^, resolution: 4 cm^−1^ and gain: 4.

The CWA degradation reaction was monitored via GC/MS method using the head-space technique after solid phase micro-extraction (SPME, Carboxen/Polydimethysiloxane Fibre, Supelco, Bellefonte, PA, USA). We used GC/MSD 7890/5975C (Agilent Technologies, Inc., Wilmington, DE, USA) equipped with an HP-5MS column (30 m length, ø of 250 μm, phase 0.25 μm) with the following parameters: 1.2 mL/min He carrier gas, 290 °C T inlet, 290 °C T interface GC/MSD, scan range 35–800 amu, 10:1 split; GC program: 40 °C for 2 min, from 40 to 280 °C at 10 °C/min and 280 °C for 10 min [9,10]. The same chemical warfare agents and the same procedure as in the FTIR measurement technique were used in the control analysis. The decreasing area of the chromatographic peak given by the chemical warfare agents relative to the original peak area was evaluated. At the same time, decomposing products of the chemical warfare agents were identified by software.

## 3. Results

### 3.1. Soman

There appeared to be a strong disposition towards reactions with nucleophilic agents on the P–F bond in the molecule *O*-(3,3-dimethyl-2-butyl)methylphosphonofluoridate. Nucleophilic fluorine substitution leads to products that are significantly less toxic, and is suitable for the decomposition of soman. Electrophilic agents are less reactive to soman. However, it is important that the reaction with the electrophilic agent results in fission of a fluoride anion; this is necessary for the decomposition of soman [31].

The P–F bond is also responsible for soman toxicity, which causes the inhibition of cholinesterase by fission of the fluoride anion. The inhibition of cholinesterase causes acetylcholine to accumulate in receptors, resulting in long-term excessive cholinergic receptor irritation. This leads to clinical outcomes involving muscarinic, nicotinic and central clinical symptoms [31].

The main issue in the study of soman decomposition is the P–F bond. In the infrared spectrum, the absorbance bands of the P–F bond vibration generally occur at wavenumbers of 1010–500 cm^−1^. From the spectra of soman mixtures with sorbents measured at different times of reaction, a band with a maximum absorbance of 840 cm^−1^ was considered optimal. An example is shown in Figure 1. The band area was between 856 and 825 cm^−1^; baseline points were the same.

When studying interfering effects, soman decomposition products were identified by the GC/MS method [9]. Decomposition products were represented by the compounds formed from a 3,3-dimethyl-2-butyl group removed from the soman molecule (predominantly alkenes, dienes, chlorinated derivatives, alcohols and ketones); bis(3,3-dimethyl-2-butyl)methylphosphonate was also identified. It is important for the process itself that there was no P–F bond substance, and thus no interference when monitoring the soman degradation.

### 3.2. Agent VX

Compared with the soman structural formula of *O*-ethyl-*S*-(diisopropylaminoethyl), the major change in the methylphosphonothioate molecule is the substitution of strongly electronegative fluorine with a far less electronegative diisopropylaminoethanthiol group. In comparison with soman, a significantly lower disposition to reactions with nucleophilic agents on the P–S bond was evident [31]. Analogous to soman, a nucleophilic substitution, the P–S bond was fissioned directly with the release of the diisopropylaminoethanethiol group. For the decomposition of VX, reactions with electrophilic agents were of considerable importance, and they did not disrupt the P–S bond directly, but they did significantly affect the subsequent reaction with nucleophilic agents. This type of reaction depends on the type of electrophilic agent [31]. In any case, the fission of the P–S bond was the reaction result.

In the infrared spectrum, absorbance bands are generally bonded by P–S–(C) bonding vibrations in compounds with a P=O bond at 613 to 510 cm^−1^. To observe VX degradation, a band with a maximum absorbance of 515 cm^−1^ was selected. An example is shown in Figure 2. The corrected band area was between 538 and 494 cm^−1^.

With regards to interference effects, it should be noted that, among the decomposition products identified by the GC/MS method [8,9], O,*S*-diethylmethylthiophosphonate, *O*-methyl-*S*-(diisopropylaminoethyl)ethylthiophosphonate and *O*-ethyl-*S*-(diisopropylaminoethyl) methyldithiophosphonate were present in traces. However, these P–S bond agents were only found in the reaction products of VX with the least effective sorbent. These may be the substances that became temporary intermediate products during the degradation, and therefore distorted measurements at the start of the reaction. Due to the minimal quantity in the mixture, only an insignificant influence on the results can be expected. The secondary and tertiary amines and diamines, thiols, sulphides, disulphides and alcohols with diisopropyl and diisopropylaminoethyl groups were identified as decomposition products in the largest amounts. Those substances do not contain a P–S bond, and thus, did not interfere with the monitoring of VX degradation.

### 3.3. Sulphur Mustard

Bis(2-chlorethyl)sulphide is simultaneously an alkyl chloride and a dialkyl sulphide. The most polar bond is between a carbon atom and a chlorine atom. A two-step reaction causes substitution of both chlorine atoms by nucleophilic reagents and production of compounds which are either slightly toxic or non-toxic (according to the nucleophile type). The fission of the C–Cl bond as a kinetic control process under normal conditions is very slow; therefore, an electrophilic attack on a sulphur atom is used to accelerate it. To monitor sulphur mustard decomposition, it is, therefore, necessary to focus on the C–Cl bond. The C–Cl bond is also the major cause of toxicity in sulphur mustard because it acts as an alkylating agent [32].

Absorbance bands corresponding to the C–Cl bond vibrations are in the range of 850–550 cm^−1^, where they are very difficult to interpret [33]. However, the sulphur mustard molecule contains CH_2_–Cl groups, which provide intensive bands in the range of 1300–1150 cm^−1^ [32]. Experimental verification detected this band for sulphur mustard at a wavenumber with a maximum absorbance of 1207 cm^−1^. The corrected area of the band was read across the wavenumber range of 1234–1198 cm^−1^, and is shown in Figure 3.

GC/MS analysis of the products of sulphur mustard degradation through decontamination sorbents [8,9] revealed that dithiane, thiirane, divinyl sulphide, oxathian, oxathiolane, thiophene and their derivatives predominated. In addition, four substances with CH_2_–Cl bonds that could potentially disturb the determination of degradation efficiency were identified.

The first was 1,2-dichloroethane. As the infrared spectra comparison in Figure 4 shows, dichloroethane did not interfere with the actual sulphur mustard measurement because the absorbance band corresponding to the CH_2_–Cl bond vibration was shifted to higher wavenumbers.

The second substance was 2-chloroethyl vinyl sulphide. It was identified only in trace amounts and only in some sorbents. It is a transient product that reacts further to divinyl sulphide. The disturbing effect of 2-chloroethyl vinyl sulphide on the overall assessment of the degradation efficiency was insignificant, and it was only observed to a small extent, at the beginning of the reaction.

Other products of the reactions with some sorbents were bis(2-chloroethyl)ether and bis(2-chloroethyl)disulphide. These are highly toxic substances, so their presence in the decomposition products indicates an insufficient ability of the sorbents to decompose sulphur mustard into non-toxic products.

## 4. Discussion

The absorbance band intensity of the CWA was proportional to the amount of substance in the sorbent mixture. Therefore, the results can be used for the quantitative expression of CWA degradation. The proportion of decomposed CWA was calculated as the relative intensity decrease in the respective band at a particular reaction time and the band intensity at the beginning of the measurement.

When verifying the measurements, it was revealed that the intensity of all absorbance bands in the spectrum gradually decreased with time. This phenomenon did not occur due to decomposition of the substance, but for other reasons, probably the gradual desorption of the agents from the sorbent and evaporation. The effect of CWA evaporation had to be eliminated.

For each CWA, absorbance bands with a high degree of stability intensity were chosen. The bands corresponding to vibrations of the −CH_3_, −CH_2_ and −CH groups were the most suitable; they were not affected by decomposition reactions. Decreases in the intensity of these bands could then be attributed only to physical phenomena and not to chemical reactions with the sorbent.

The intensity of the bands corresponding to the monitored bonds were then related to the band intensity, corresponding to the vibrations of mentioned groups (so-called reference absorbance band). Therefore, the ratio of the corrected areas of those stable bands and the monitored CWA bands was used for the actual evaluation. The reference band wavenumber ranges are shown in Table 1.

The following procedure is a result of the optimisation of monitoring CWA decomposition using decontamination sorbents:Measurement of background and sorbent mixture with CWA at selected time intervals is done according to Section 2. The ratio of the amount of CWA and the amount of sorbent can range, according to the requirements, between 1:20 and 1:10;For all measured spectra, the areas of monitored bands (A_m_) and reference bands (A_ref_) are subtracted in the wavenumber range, according to Table 1;For each time, the area ratio Y_m_ = A_m_/A_ref_ is calculated. For a spectrum corresponding to 0 min, the ratio is denoted as Y_0_ = A_0_/A_ref_;The relative part of decomposed CWA (RPD) is calculated from the equation: RPD (%) = (Y_0_–Y_m_) × 100/Y_0_;Time dependency of the relative part of the decomposed CWA is formed.

A concrete example of the processing of the results for the soman decomposition measurement of the decontamination sorbent LB3 is given in Table 2; the corresponding graph of time dependency is shown in Figure 5.

The time dependence allowed the description of the degradation efficiency of sorbents by quantitative characteristics. For these purposes, the times required for 50% (t_50_) and 90% (t_90_) of CWA decomposition were expressed. These times were determined by extrapolation from graphical relationships, as shown in Figure 5. In this way, the set experimental conditions allowed the expression of differences between the different decontamination sorbents and the characterisation of their degradation efficiencies.

Decomposition of CWA by decontamination sorbents was verified using the GC/MS method; the results fully correlated with the results of the FTIR measurements.

## 5. Kinetics of Decomposition

The dependencies, according to Figure 5, were instrumental in the simple and rapid determination of CWA degradation using sorbent by quantitative characteristics t_50_ and t_90_. The measured data also enabled the kinetics of decomposition to be identified.

In all experiments, a CWA reacted with sorbent, which was in excess. As the concentrations of sorbent did not change much during the process, we considered them constant. In the rate equation, the concentration of sorbent was considered constant and its concentration was included in the rate constant. Thus, the order of reaction is now one. It is, therefore, a pseudo-first-order reaction, which is described by the first-order reaction equation [34].

Assuming the hypothesis that decomposition proceeds as a pseudo-first-order reaction, the general equation C_t_ = C_0_ × e ^(−k^**^’^**^t)^ can be applied, where C_t_ is the amount of CWA at time t, C_0_ is the initial amount of CWA and k’ is a reaction rate constant, which also includes an amount of sorbent which is almost constant during decomposition. The significance of RPD values in percentage units indicates that C_t_ is the difference between 100 − RPD and C_0_ equals 100. The logarithmic form of the first-order reaction equation is then
ln(100/(100 − RPD)) = k × t

The dependency of ln(100/(100-RPD)) on time is linear for all CWAs and all sorbents. The dependencies are shown in the example of the decomposition of VX for a CWA/sorbent weight ratio of 1:10, in Figure 6. This is clear proof of the hypothesis that CWA decomposition using tested sorbents follows the first-order equation [34].

The slope of the linear dependencies in Figure 6 represents a reaction rate constant k’, which, in addition to the constant k, also includes the amount of sorbent. Linear dependencies can be also used to calculate the times required for 50% (t_50_) and 90% (t_90_) CWA decomposition. Those times are equal to: t_50_ = ln2/k’, t_90_ = ln10/k’. The values of the reaction rate constant k’ and times t_50_ and t_90_ for individual CWA and sorbents are shown in Table 3.

The accuracy of determined times t_50_ and t_90_ was defined during the process validation. The values in Table 3 represent the average values collected from five tests. The accuracy of time determination is expressed by a relative standard deviation s_r_.

The results show that the rate of CWA decomposition due to decontamination sorbents decreased in the order of soman–agent VX–sulfur mustard. The evaluation also showed that, under the given conditions, only FAST-ACT and LB3 could degrade more than 90% of all CWAs within two hours.

The overall accuracy of the time determination, expressed as a relative standard deviation, was 11% for t_50_ and 8% for t_90_, respectively. The determination of low t_50_ values was the least accurate in cases of very rapid decomposition of CWA. The relative standard deviation was close to 20%.

To assess the accuracy of the t_50_ and t_90_ determinations, an interlaboratory comparison of VX degradation testing of sorbents LB1, LB3 and LB4 was carried out. Five laboratories of the Fire Rescue Service of the Czech Republic perfomed the tests. They carried out three tests for each sorbent. The results are shown in Table 4. Considering the designation and objectives of the decontamination sorbent testing methodology, the precision of the t_50_ and t_90_ times was assessed to be sufficient.

## 6. Conclusions

The infrared spectra of CWA mixtures with solid decontamination sorbents, acquired using the ATR technique, were studied. Measuring the intensities of characteristic absorbance bands was used to objectively evaluate the degradation efficiency of solid decontaminants. The designed procedure is based on the measurement of the infrared spectra at certain time intervals and the utilisation of the intensities of the characteristic CWA absorbance bands to quantitate their loss.

The proportions of degradation products of chemical warfare agents as a function of time, and time required for the decomposition of 50% and 90% of the starting substance quantity, formed the results of the measurement.

The described test procedure allowed the FRS chemical laboratories to assess whether the decontamination sorbents with which the FRS units are equipped are sufficiently effective and meet the relevant technical conditions. This work paves the way for continued development of procedures for monitoring the decontamination of other CWAs and highly toxic substances.

## Figures and Tables

**Figure 1 toxics-07-00063-f001:**
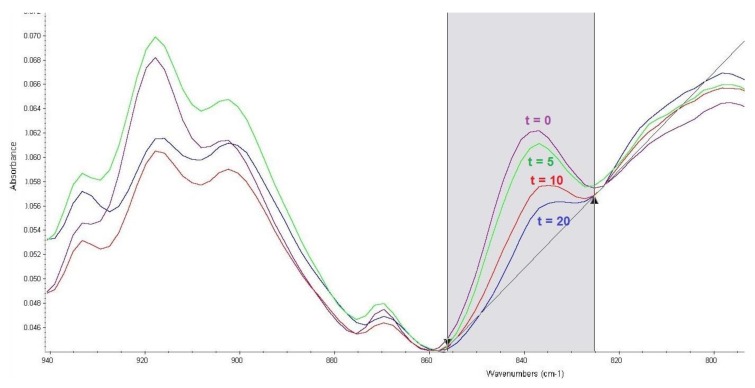
Effect of reaction time (t, min) of soman with FAST-ACT sorbent on an absorbance band corresponding to P–F bond vibrations.

**Figure 2 toxics-07-00063-f002:**
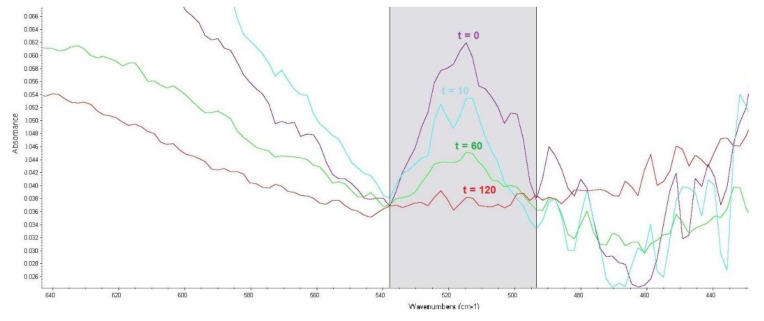
Effect of reaction time (t, min) of VX with LB4 sorbent on an absorbance band corresponding to P–S–(C) bond vibrations.

**Figure 3 toxics-07-00063-f003:**
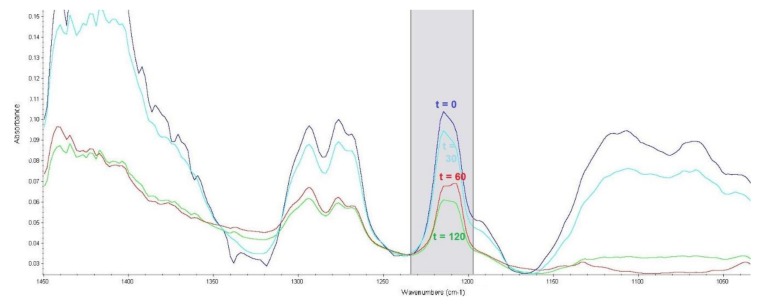
Effect of reaction time (t, min) of sulphur mustard with the LB3 sorbent on an absorbance band corresponding to CH_2_–Cl bond vibrations.

**Figure 4 toxics-07-00063-f004:**
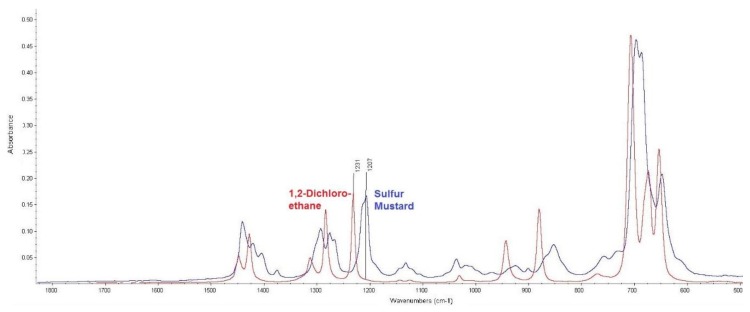
Comparison of infrared spectra of sulphur mustard and 1,2-dichloroethane (measured by attenuated total reflection (ATR) technique).

**Figure 5 toxics-07-00063-f005:**
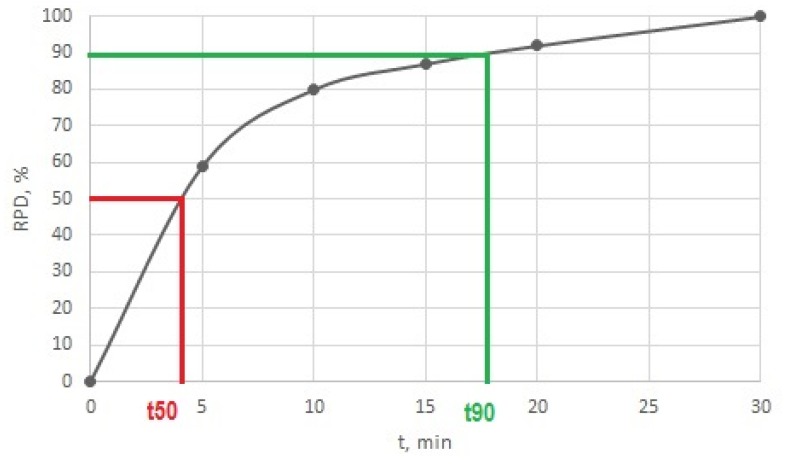
Relative part of decomposed soman (RPD) by decontamination sorbent LB3 (CWA/sorbent weight ratio is 1:5) as a function of time.

**Figure 6 toxics-07-00063-f006:**
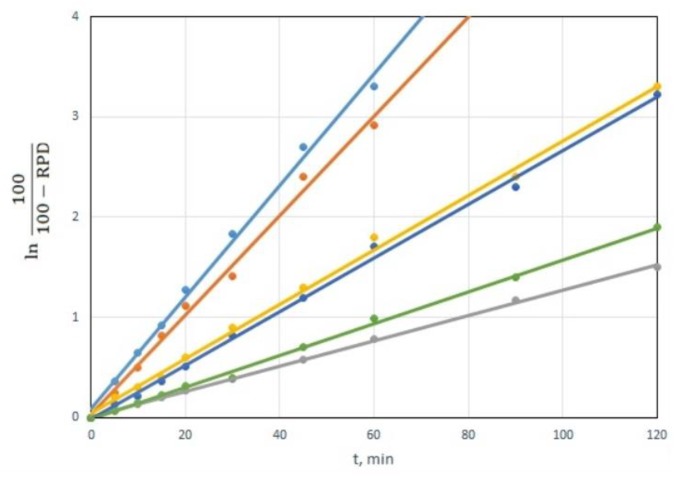
Dependence of value ln(100/(100-RPD)) on time for VX degradation by decontamination sorbents (CWA/sorbent weight ratio 1:10).

**Table 1 toxics-07-00063-t001:** Wavenumber ranges that were to used determine the areas of absorbance bands and baseline points.

CWA	Soman	VX	Sulphur Mustard
Monitored band, cm^−1^	856–825	538–494	1234–1198
Reference band, cm^−1^	3012–2887	990–910	1320–1250

**Table 2 toxics-07-00063-t002:** Results of monitoring the soman decomposition process using decontamination sorbent LB3.

t, min	0	5	10	15	20	30
A_m_ (856–825 cm^−1^)	0.42	0.15	0.069	0.035	0.021	0
A_ref_ (3012–2887 cm^−1^)	1.15	0.97	0.93	0.74	0.69	0.63
Y_m_ = A_m_/A_ref_	0.37	0.15	0.074	0.047	0.030	0
The relative part of decomposed CWA RPD, %	0	59	80	87	92	100

**Table 3 toxics-07-00063-t003:** Reaction rate constants k’, times t_50_, t_90_ and relative standard deviations s_r_ for decomposition of chemical warfare agents (CWAs) by decontamination sorbents (number of measurements, five; CWA/sorbent weight ratio 1:10).

CWA	Sorbent	k’, min^−1^	t_50_, min	s_r_, %	t_90_, min	s_r_, %
Soman	FAST-ACT	0.10	6.9	18.1	23	10.4
	LB1	0.021	33	10.3	110	4.9
	LB2	0.0026	>120	-	>120	-
	LB3	0.16	4.3	19.8	14	13.2
	LB4	0.039	18	12.7	59	6.6
	LB5	0.024	29	9.5	96	5.8
Agent VX	FAST-ACT	0.056	12	14.7	41	8.0
	LB1	0.049	14	15.5	47	8.6
	LB2	0.013	53	11.3	>120	-
	LB3	0.025	28	8.4	92	7.3
	LB4	0.027	26	7.6	85	5.9
	LB5	0.016	43	5.7	>120	-
Sulphur	FAST-ACT	0.025	28	9.1	92	5.2
mustard	LB1	0.010	69	7.4	>120	-
	LB2	0.0051	>120	-	>120	-
	LB3	0.021	33	11.0	110	7.9
	LB4	0.0019	>120	-	>120	-
	LB5	0.0089	78	6.8	>120	-

**Table 4 toxics-07-00063-t004:** Results of interlaboratory comparison of t_50_ and t_90_ degradation tests of VX by decontamination sorbents (CWA/sorbent weight ratio 1:10).

Sorbent	LB1	LB3	LB4
Total number of results	15	15	15
Number of results out of tolerance ±20%	1	0	0
Average interlaboratory value t_50_ (min)	12	28	28
Relative standard deviation t_50_ interlaboratory (%)	10.3	8.6	9.0
Average interlaboratory value t_90_ (min)	42	97	93
Relative standard deviation t_90_ interlaboratory (%)	9.0	9.3	8.6

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
