# Peer review of "Study of Decomposition of Chemical Warfare Agents using Solid Decontamination Substances"

_toxics, 2019, doi:10.3390/toxics7040063_

Round 1
Reviewer 1 Report
I now recommend publication of this edited version.
Author Response
Thank you for your recommendations of the publication.
Tomas Capoun
Reviewer 2 Report
The revised manuscript addressed some of the concerns that I have raised previously however I am still having hard time understand the novelty of studying a commercially available material for CWA degradation which is designed for the same application.
Author Response
Point: I am still having hard time understanding the novelty of studying a commercially available material for CWA degradation which is designed for the same application.
Response: Commercial materials are verified by the manufacturer and designated institutions. However, each country has different requirements for decontamination efficiency, different assumed default contamination densities and sorbent consumption standards. The novelty is that, according to the published work, each customer can verify in a simple and reproducible manner that the sorbent meets their specific requirements. This is stated at the end of the thesis.
Tomas Capoun
Reviewer 3 Report
This contribution from Capoun & Krykorkova describes the use of FTIR spectroscopy to follow the decontamination kinetics of chemical warfare agents VX, GD and HD on solid sorbent decontaminants/materials. FTIR is targeted as an analytical technique because of its ease-of-use and common availability within labs. The authors focus on using distinct FTIR signals that are present in the intact CWA but not present in the decontaminated agent in order to quantify the extent of decontamination. The analysis method gives consistent results and shows lab-to-lab and operator-to-operator reproducibility.
The interpretation of FTIR spectra and extrapolation of this data to quantify decontamination requires more support:
The authors should include a calibration curve taken with sorbent doped with different amounts of agent in order to demonstrate that the FTIR peak areas consistently correlate with the amount of agent present. In order to conclude that changes in FTIR spectra are due to CWA decontamination, the authors need to verify the decontamination efficacy of the materials using another analytical method such as extraction of the sorbent followed by GC analysis, solid-state NMR, etc. This is necessary in order to definitively state that the decreases in FTIR peak intensity are indeed due to covalent bond breakage leading to decontamination/neutralization of the CWA. Adsorption can lead to loss of FTIR peaks when the corresponding bond vibration/stretch is suppressed through non-covalent interaction with the sorbent material - in this situation, the agent is not decontaminated or neutralized, and the contaminated sorbent material still presents a health hazard. Are there changes in the FTIR spectra that indicate the formation of expected decontamination products? If so, these changes should be discussed.An important limitation of this work is that the characteristic FTIR signals of each agent needs to be assigned in order to know if the specific peak is characteristic of a functional group that is irreversibly changed in the decontamination/neutralization reaction. For HD, not all the IR peaks are unambiguously assigned (10.1002/jrs.1250220807, 10.1021/j100339a018, 10.1021/la981440v), specifically with respect to the 1234-1198 cm-1 band attributed here to CH2-Cl stretch. This makes it even more crucial that the efficacy of decontamination by adsorbent materials reported in this manuscript be verified by a different analytical method. The authors should also note that different decontaminants react with CWAs differently - although metal oxide-based sorbents like Fast-Act (TiO2/MgO mix) degrade agents in a nucleophilic manner, another common solid decontaminant BX-24 contains electrophilic chlorine and degrades CWAs in an oxidative manner. The IR analysis method reported in this manuscript would not be applicable to non-nucleophilic decontaminants.
If the decontamination efficacy results derived from FTIR measurements reported in this manuscript can be quantitatively correlated to decontamination efficacy of the same materials derived from another analytical method, then the concept of a simple and highly reproducible method for evaluating solid decontaminants presented here will be an important finding that is of interested to the chemical hazard decontamination community at large.
Author Response
Please see the attachment.
Tomas Capoun

Round 2
Reviewer 3 Report
Addition of GC-MS analysis details greatly increased the credibility of this study. Again, the concept presented in the paper of a simple, user-friendly method for non-experts to test commercially available solid-phase decontamination materials is important. By putting the FTIR results alongside GC-MS results allows readers to have confidence in the science presented in this paper and therefore adopt similar methods in their own agencies/labs.
There are remaining grammar and syntax errors that need to be revised for ease of reading and unambiguous interpretation of meaning. The manuscript should be copy-edited by someone fluent in English and chemical jargon. A frequent error in this manuscript is the use of 'binding' rather than 'bond'. The term 'binding' in chemistry tends to refer to adsorption, chelating, or dative interactions, while 'bond' unambiguously refers to chemical bonds (usually covalent or ionic). For example, in line 153/154, the sentence reads better if written as "In any case, fission of the P-S bond is the reaction result."
Author Response
Dear Reviewer. 
Thank you for reviewing our manuscript. Your comments have been acknowledged. The manuscript was proofread by our native English speaking colleague – a chemist. To remove grammatical errors, the manuscript was submitted to the MDPI English editing service. 
Yours sincerely 
Tomas Capoun 
Jana Krykorkova
This manuscript is a resubmission of an earlier submission. The following is a list of the peer review reports and author responses from that submission.
Round 1
Reviewer 1 Report
Current work describes the decontamination of chemical warfare agents using commercially available solid sorbents and monitor the reactivity using IR spectroscopy. Considering that the studied solid is a commercially available solid sorbent designed for the same purpose studied here, it is not clear to me what is the novelty of the current work. Authors should clearly explain the impact of the work. After the following points are addressed it should be reconsidered for publication.
1- Since FASTACT is a commercial product designed for the same purpose, isnt the company who makes it study all these properties?
2- Is there any conflict of interest between the company and researchers in this work?
3- What is the composition and structure of the sorbent used? How does it interact and decompose the agents?
4- Since the IR has been directly used for quantification of the reactivity, there needs to be stadard curve with know amounts of the reagent to show the viability of the method
5- The conversion profile shown in the last figure should be fitted 1st, 2nd order or another equation for determination of the reaction rate.
Author Response
Please see the attachment.
Tomas Capoun

Reviewer 2 Report
The authors report the decomposition of chemical warfare agents (CWAs), both nerve agents (GD and VX) and sulfur mustard using vibrational spectroscopy (FTIR/ATR) along with some guidelines so that these techniques can be used generally to assess decontamination of the CWAs by active solids in the future. On the positive side, a general and inexpensive method for quantification CWA decontamination by solids is needed, and the method outlined here is certainly easily executed and inexpensive. However, the FTIR/ATR, a very well developed and used technique (no novelty here) is hard pressed to be a quantitative measure of reactions on solids because the resolution and overlap of peaks (reactant and products) is a problem (FTIR/ATR is simply not a very high resolution method; Raman is better suited but admittedly there are fewer Raman instruments available and they remain somewhat more expensive.) Regarding novelty (or the lack thereof), the authors state, quoting (page 2): “The methodology for evaluating the degradation efficiency of solid decontamination substances has so far been non-existent.” This isn’t really true. The Klabunde laboratory and NanoScale who developed Fast Act (nanoscale MgO, TiO2) and related reactive decontaminating solids used solid state NMR to follow CWA decon on various solids (31P CP MAS for nerve agents and 13C CP MAS for mustard and 27Al CP MAS NMR for Al2O3 powder decontaminants: see for example J. Phys. Chem. B 2000, 104, 5118-5123 and J. Am. Chem. Soc. 2001, 123, 1636-1644), and other groups routinely wash spent decon solids and assess the degree of CWA removal and quantify the products by GC or GC/MS. These authors do use some other methods that are specific for various CWAs, techniques that others have used I believe.
Weaknesses in the work include the fact that despite the authors’ identification of several degradation products of GD, VX and HD, the ratios of these products aren’t given. The product distribution provides information on the nature and effectiveness of the degradation. The kinetics (e.g. Figure 5) are not highly informative: (a) there is no interpretation of the kinetics as it relates the actual reactions (what processes are involved as the CWAs interact with the solid decontaminants). (b) More data points would facilitate a better interpretation of the reactions, and (c) the extrapolation to 100% yield is faulty. The authors show a straight line going to completion (100% reaction). However, these reactions, like the nearly all solution as well as liquid-solid interfacial reactions, are positive order overall (pseudo-first-order or other orders) and thus do not go to completion. In other words, the kinetics are not a straight line (zero-order kinetics) but curve and never reach 100% completion (the asymptote of the correct reaction function with is a rectangular hyperbolic function).
The authors state, quoting: “When verifying the measurements, it was revealed that the intensity of all absorbance bands in the spectrum gradually decreases with time. This phenomenon does not occur due to decomposition of the substance but for other reasons, probably by gradual desorption of the agents from the sorbent and by evaporation. The effect of CWA evaporation had to be eliminated.” Errors due to both unwanted sorption and evaporation should be measured, quantified and described this way in the text. They shouldn’t be the source of speculation.
There are many studies involving solid CWA decontaminating materials (MOFs, POMs and other sorbents) that aren’t footnoted, including 3 (and possibly 4) reviews in Chem Reviews. These should be footnoted in a revised ms.
In summary, the main method proposed and addressed in this ms isn’t very convincing for quantification of CWA decon on solids. FTIR/ATR just isn’t a high resolution technique – the bands of several products overlap too severely. The counterpoint is that there isn’t any method for quantification of CWA decon on solids that is also quick and inexpensive. Solid state NMR or a combination of methods are much more satisfactory but admittedly, SS NMR isn’t readily available to everyone. Several aspects of this study are somewhat naïve and incomplete. As a result, I would argue that the authors do more work (collect more data and analyze it more effectively taking into account the above issues) and re-submit a stronger manuscript.
Minor:
The English needs some improvement – mostly poor sentence structure and the use of articles.
Author Response
Please see the attachment.
Tomas Capoun

Round 2
Reviewer 1 Report
The revised version is suitable for publication.
Author Response
Dear Reviewer,
Thank you very much for your comments on our paper.
Yours sincerely
Jana Krykorkova, Tomas Capoun
Reviewer 2 Report
Journal
Toxics (ISSN 2305-6304)
Manuscript ID toxics-577977
Type Article
Number of Pages 8
Title: Study of Decomposition of Chemical Warfare Agents using Solid Decontamination Substances
Authors: Tomas Capoun , Jana Krykorkova *
The authors’ reply to the first 2 critiques (reviewer’s responses) and critique number 6 help clarify that this work is more relevant to internal use in the Czech republic rather than general international protocols for decontamination solids. They haven’t attempted to make this study more broadly applicable and haven’t cited more informative work by several other labs. They have ignored multiple suggestions that would put this work on a more informative footing. Nonetheless, if the authors are happy with this, then I suggest publication. Some specific comments:
Response 3 – both rates and products are important.
Response 4 – OK.
Response 5 – The reviewer suggestions ignored.
Author Response
Dear Reviewer,
Thank you very much for your comments on our paper.
We would like to address your remark concerning the ignored proposal that: “There are many studies on solid CWA decontamination materials (MOF, POM and other sorbents) that are not footnoted, including 3 (and possibly 4) reviews in Chem reviews. These should be footnoted in a revised ms”:
Links to some studies related to the topic have now been added to the paper (line 65-69, 305-353).
Yours sincerely
Jana Krykorkova, Tomas Capoun